# Genomic Evolution of ST11 Carbapenem-Resistant *Klebsiella pneumoniae* from 2011 to 2020 Based on Data from the Pathosystems Resource Integration Center

**DOI:** 10.3390/genes13091624

**Published:** 2022-09-10

**Authors:** Na Zhang, Yue Tang, Xiaojing Yang, Meiling Jin, Jiali Chen, Shiyu Qin, Fangni Liu, Xiong Liu, Jinpeng Guo, Changjun Wang, Yong Chen

**Affiliations:** 1School of Public Health, China Medical University, Shenyang 110122, China; 2Department of Emergency Response, Chinese PLA Center for Disease Control and Prevention, Beijing 100071, China; 3College of Public Health, Zhengzhou University, Zhengzhou 450001, China; 4Department of Information, Chinese PLA Center for Disease Control and Prevention, Beijing 100071, China

**Keywords:** global, whole-genome sequence, serotype, distribution characteristics, evolution

## Abstract

(1) Objective: ST11 carbapenem-resistant *Klebsiella pneumoniae* (CRKP) is widespread throughout the world, and the mechanisms for the transmission and evolution of major serotypes, ST11-KL47 and ST11-KL64, were analyzed to investigate the global distribution and evolutionary characteristics of ST11 CRKP; (2) Methods: The Pathosystems Resource Integration Center (PATRIC) database was downloaded and all *K. pneumoniae* from 2011 to 2020 were screened to obtain ST11 CRKP genome assemblies with basic information. The relationship of serotype evolution between KL47 and KL64 was then investigated using statistical and bioinformatic analysis; (3) Results: In total, 386 ST11 CRKP isolates were included for analysis. Blood (31.09%, 120/386), respiratory tract (23.06%, 89/386), and feces (20.21%, 78/386) were the major sources of samples. China was the leading country where ST11 CRKP was isolated. KL47 and KL64 were found to be the most prevalent serotypes. ST11-KL64 CRKP [median 78(*P*_25_~*P*_75_: 72~79.25)] had remarkably more virulence genes than the KL47 [median 63(*P*_25_~*P*_75_: 63~69)], and the distinction was statistically significant (*p* < 0.001). A differential comparison of virulence genes between KL47 and KL64 revealed 35 differential virulence genes, including *rmpA*/*rmpA2*, *iucABCD*, *iutA*, etc. The comparison of the recombination of serotype-determining regions between the two serotypes revealed that KL64 CRKP carried more nucleotide sequences in the CD1-VR2-CD2 region than KL47 CRKP. More nucleotide sequences added approximately 303 base pairs (bp) with higher GC content (58.14%), which might facilitate the evolution of the serotype toward KL64; (4) Conclusions: KL47 and KL64 have become the predominant serotypes of ST11 CRKP. KL64 CRKP carries more virulence genes than KL47 and has increased by approximately 303 bp through recombinant mutations, thus facilitating the evolution of KL47 to KL64. Stricter infection prevention and control measures should be developed to deal with the epidemic transmission of ST11-KL64 CRKP.

## 1. Introduction

As an important Gram-negative bacterium, *Klebsiella pneumoniae* can cause a variety of healthcare-associated infections, including pneumonia, bloodstream infections, and wound or surgical site infections. The frequent use of antibiotics has made it easier for *K. pneumoniae* to undergo chromosomal changes and develop antibiotic resistance [1]. Carbapenem-resistant *K. pneumoniae* (CRKP) belongs to a commonly detected pathogen in hospitals all around the world. CRKP infection has become a serious health threat because of its aggressive pathogenesis [2], poor prognosis, and high mortality [3,4].

Multilocus sequence typing (MLST) uses nucleic acid sequences to classify bacteria and examine strain diversity by amplifying the internal segments of seven housekeeping genes and analyzing their sequences. Serotyping is an immunological technique that can classify the same bacterium into different subspecies and is used to distinguish several forms of the same pathogen. The predominant multilocus sequence type in China is ST11, while ST258 is the predominant sequence type in the United States and European countries [5,6]. Both ST11 and ST258 belong to the CG258 clonal lineage. The most common and effective method for serotyping is to compare the DNA homology of the *wzc* gene’s CD1-VR2-CD2 variable region. In this region, 80.00% of homology is considered different serotypes, while more than 96.00% of homology is considered the same serotype [7]. In recent years, the proportion of ST11-KL64 in CRKP strains has gradually increased [8], while ST11-KL47 remains dominant in some areas [9].

Some studies have suggested that serotypes ST11-KL47 and ST11-KL64 have an evolutionary relationship, and recombinant mutations have occurred in ST11-KL47 strains, resulting in serotype conversion to KL64 [10]. ST11-KL64 CRKP carried the virulence genes (*rmpA*/*rmpA2*, *iucABCD* and *iutA*), and the presence of these virulence genes could be linked to the transformation of KL47 into KL64.

To investigate the genomic differences and evolutionary mechanisms between the two serotypes, we performed a study based on screening and analyzing ST11 CRKP genomic data sourced from the Pathosystems Resource Integration Center (PATRIC) [11] database between 2011 to 2020.

## 2. Materials and Methods

### 2.1. Data Sources

All *K. pneumoniae* genome sequences from 2011 to 2020 were collected from the PATRIC database, which was screened based on the host, sample sources, presence of the carbapenemase-encoding gene, and isolation nation. A human host, eight sample sources (blood, urine, feces, bronchoalveolar lavage, respiratory secretions, wound pus, catheter, and sterile body fluids), and carriage of any one of the carbapenemase-encoding genes (including *bla*_KPC_, *bla*_NDM_, *bla*_VIM_, *bla*_IMP_, etc.) were used as inclusion criteria. As the number of KL47 and KL64 serotypes did not change significantly over a short period of time, such as one year, whereas the change in the dominance of KL47 and KL64 serotypes in ST11 CRKP over five years was more intuitive, and thus the samples were divided into two periods for comparison: 2011–2015 and 2016–2020.

### 2.2. Analysis of Virulence Genes, Resistance Genes, and Plasmids

The strains were identified and screened for carrying carbapenemase-encoding genes using Kleborate [12] software. The number of virulence genes, resistance genes, and plasmids carried by each strain were determined by comparing with CARD [13], VFDB [14], and Plasmid Finder [15] databases using Abricate software, and genome annotation was performed using Prokka v1.12 [16].

### 2.3. Bioinformatics Analysis

Core genome alignment was obtained using Roray v3.13.0 [17], single nucleotide polymorphism (SNP) recombination analysis was performed using snippy, the recombination results were combined and the best model was calculated using modeltest-ng, while the maximum likelihood tree was drawn using raxml-ng afterward, and the phylogenetic tree obtained by the maximum likelihood method was used along with the recombination data using ClonalFrameML v1.12 [18] to deconstruct and obtain the reconstructed images. The phylogenetic tree was displayed using iqtree v2.2.0.3 [19]. Jalview v2.11.2.2 [20,21] was used for alignment display, and a comparison of KL47 and KL64 representative strains with *wzc* genes was performed using blast v2.13.0, and visualized by CGview [22].

### 2.4. Differential Gene Analysis

GO enrichment analysis of virulence genes with statistically significant differences in the number of virulence genes between KL47 and KL64 CRKP was performed using DAVID [23] software, and bubble plots were generated using R version 4.2.0 (R Core Team. R: A language and environment for statistical computing. R Foundation for Statistical Computing, Vienna, Austria; 2022. URL: https://www.R-project.org/ (accessed on 10 May 2022)).

### 2.5. Statistical Analysis

Statistical analysis was performed using R version 4.2.0 (R Core Team. R: A language and environment for statistical computing. R Foundation for Statistical Computing, Vienna, Austria; 2022. URL: https://www.R-project.org/ (accessed on 10 May 2022)). Firstly, the numbers of virulence genes, resistance genes, and plasmids carried by KL47 and KL64 CRKP were compared. Continuous variables that met normality were described using the mean ± standard deviation, and differences between the two groups were compared using the two independent samples *t*-test. If continuous variables did not meet normality, they were described using the median (interquartile), and non-parametric tests (Mann–Whitney-U test) were used to compare differences between groups. The chi-square test of independence was used to compare the differences in the proportions of resistance or virulence genes in different groups. When one or more cell counts in a 2 × 2 table were less than 5, Fisher’s exact test was used. A *p*-value < 0.05 was considered statistically significant.

## 3. Results

### 3.1. Clinical and Molecular Characterizations of ST11 CRKP

There were 2356 CRKP genome assemblies obtained after database screening, including 1620 CRKPs from 2011 to 2015, and 736 CRKPs from 2016 to 2020. There were 165 (10.19%, 165/1620) ST11 CRKP genome assemblies in 2011–2015, and 221 (30.03%, 221/736) ST11 CRKP genome assemblies in 2016–2020, the information of 386 ST11 CRKP strains was in Appendix A. In total, blood samples accounted for 31.09% (120/386) of all sample sources, while the proportions of respiratory fluids, feces, urine, wound pus, alveolar lavage fluid, sterile body fluids, and catheter samples were 23.06% (89/386), 20.21% (78/386), 18.39% (71/386), 2.85% (11/386), 2.07% (8/386), 1.30% (5/386), and 1.04% (4/386), respectively. Between 2011 and 2015, the top three carbapenemase genes were *bla*_KPC-2_ (84.24%, 139/165), *bla*_NDM-1_ (6.06%, 10/165), and *bla*_OXA-48_ (3.03%, 5/165). From 2016 to 2020, the top three carbapenemase genes were *bla*_KPC-2_ (81.45%, 180/221), *bla*_NDM-1_ (8.60%, 19/221), and *bla*_OXA-48_ (4.52%, 10/221). In total, 386 ST11 CRKP strains belonged to 51 serotypes, including 20 serotypes during 2011–2015, and 36 serotypes during 2016–2020. KL47 was the predominated serotype (44.85%, 74/165) and KL64 made up 10.91% (18/165) of all serotypes in the first five years. During 2016–2020, KL64 accounted for 47.06% (104/221), while KL47 accounted for 16.74% (37/221) of all serotypes. (Figure 1).

There were 111 KL47 CRKP, and the proportions of blood samples, respiratory secretions, urine, feces, wound pus, bronchoalveolar lavage, catheter, and sterile body fluids were 31.53% (35/111), 25.23% (28/111), 18.92% (21/111), 16.22% (18/111), 3.60% (4/111), 1.80% (2/111), 1.80% (2/111), and 0.90% (1/111), respectively. The main country sources of KL47 were China (59.46%, 66/111) and Brazil (9.91%, 11/111), and the highest numbers were from the years 2016 (27.03%, 30/111) and 2018 (20.72%, 23/111). There were 122 KL64 CRKP, and the proportions of blood, feces, respiratory secretions, urine, catheter, sterile bodily fluids, wound pus, and bronchoalveolar lavage were 36.89% (45/122), 27.87% (34/122), 17.21% (21/122), 14.75% (18/122), 1.64% (2/122), 0.82% (1/122), 0.82% (1/122), and 0, respectively. The majority of the strains collected from 2011 to 2020 were from China (64.51%, 249/386), Brazil (9.84%, 38/386), and the United States (8.55%, 33/386), with the most strains being collected in the years 2015 (18.13%, 70/386) and 2016 (23.58%, 91/386).

### 3.2. Presence of Carbapenem Resistance and Virulence Genes in KL47 and KL64

A total of five kinds of carbapenemase genes were identified in the KL47 CRKP strains, including *bla*_KPC-2_ (96.40%, 107/111), *bla*_NDM-1_ (1.80%, 2/111), *bla*_NDM-5_ (0.90%, 1/111), *bla*_VIM-1_ (0.90%, 1/111), and *bla*_OXA-245_ (0.90%, 1/111). Thirty-four KL47 CRKP strains carried the virulence genes *iucABCD*/*iutA*, fifteen KL47 CRKP strains carried *rmpA* genes and thirty-three KL47 CRKP strains carried *rmpA2* genes. Four carbapenemase genes were present in the KL64 CRKP strains, including *bla*_KPC-2_ (97.54%, 119/122), *bla*_OXA-181_ (1.64%, 2/122), *bla*_OXA-48_ (0.82%, 1/122), and *bla*_KPC-30_ (0.82%, 1/122). A total of 50.00% (61/122) of KL64 CRKP strains carried *iucCD*/*iutA*, while 50.82% (62/122) of KL64 CRKP carried *iucAB*. The KL64 and KL47 carried equivalent numbers of resistance genes [median 15 (*P*_25_~*P*_75_: 14~18)]. However, the ST11-KL64 CRKP [median 78(*P*_25_~*P*_75_: 72~79.25)] had remarkably more virulence genes than the ST11-KL47 CRKP strains [median 63(*P*_25_~*P*_75_: 63~69)] (*p* < 0.001). In comparison to ST11-KL64 CRKP [median 3(*P*_25_~*P*_75_: 3~4)], ST11-KL47 CRKP carried more plasmids [median 4(*P*_25_~*P*_75_: 3~4)] (*p* < 0.003) (Table 1).

### 3.3. Comparison of the Virulence Genes Distribution between KL47 and KL64

To determine the differences in virulence genes carried by KL47 and KL64, we examined the 134 virulence genes carried by both ST11-KL47 and ST11-KL64 in the VFDB database. As a result, the proportions of 35 virulence genes were significantly different between the KL47 and KL64 strains (*p* < 0.05) (Table 2). The *wbbM*, *wbbN*, *wbbO*, *wzm*, and *wzt* genes are lipopolysaccharide-related genes that play a role in O-antigen processing, lipopolysaccharide synthesis, and virulence evolution. All the ST11-KL47 strains carry *the entF* gene, while 92.62% (113/122) of the ST11-KL64 strains carry this gene. More ST11-KL64 strains carry the *glf* (95.90%, 117/122) and *gnd* (100%, 122/122) genes. The *entF* gene is a cytoplasmic enzyme for making siderophore enterobactin. The *glf* gene is a capsule biosynthesis gene that is essential for capsule biosynthesis. The *gnd* gene is a gluconate dehydrogenase gene and has a highly variable nucleotide sequence that plays a facilitating role in virulence evolution. The carriage of *gnd* and *glf* in ST11-KL64 strains might play an important role in the evolution of virulence.

Fourteen genes were discovered to be enriched in five pathways after the aforementioned thirty-five differential genes were subjected to GO enrichment analyses. In the GO:0019290 pathway (Figure 2), which contains the chemical reaction and process leading to the creation of iron carriers produced by aerobic or parthenogenic anaerobic bacteria with low molecular weight Fe (III) integrators, the iron carrier genes *iucABCD* and *iutA* are primarily enriched. *Clb* is an ICEKp-associated pathogenicity locus that is enriched in three pathways, including transferase activity, transferring acyl groups, phosphopantetheine binding, and ligase activity. The *iro*, *iuc*, *rmpA*, and *rmpA2* are loci associated with pathogenic plasmids, and KL64 CRKP strains carry more genes, including *Clb*, *iro*, *iuc*, *rmpA*, and *rmpA2* genes, the difference being statistically significant. It is hypothesized that ST11-KL64 CRKP contains more virulence genes than KL47 due to iron carrier-associated chemical reactions that promote bacterial growth, reproduction, and virulence spread, as well as the pathogenicity locus gene *Clb*, which plays an important role in virulence evolution. Some studies have confirmed that these genes in ST11-KL64 strains arose during recombination due to the gain or loss of gene clusters involved in heavy metal resistance and mobile genetic elements, as a manifestation of the evolution of ST11-KL47 to ST11-KL64 after a recombination event [10].

### 3.4. Serotype Evolutionary Characterization of ST11 CRKP

All the 386 ST11 CRKP strains can be divided into 9 clades in the phylogenetic tree. KL125 belonged to the original evolutionary serotype of ST11 strains, followed by KL14 and KL24; then, KL105 and KL27 underwent evolutionary mutations to evolve into the currently popular serotypes, including KL64, KL47, K21, etc. Most of these unknown serotypes shared sequence similarities with KL107, indicating that KL107 represents an intermediate type from which several strains have evolved. China submitted 249 CRKP strains with 33 serotypes and the maximum number of CRKP strains (64.25%, 248/386), according to an analysis of the phylogeny of ST11-CRKP strains from different nations around the world (Figure 3). Strains from the same period and geographical area show more homology in phylogenetic branches. Some CRKP strains from different regions also show a high degree of evolutionary homology, suggesting that the evolution of CRKP strains needs to be considered from a global perspective.

Our analysis of SNP recombination events between ST11 CRKP strains revealed that KL47 and KL64 in the same large clade were highly similar in the locations where recombination occurred (Figure 4). In contrast, KL64 had only two more recombination regions (249514-250776, 1119662-1120610) than KL47, which carried 6 and 5 SNPs, respectively.

To verify the relationship between KL47 and KL64, we drew a phylogenetic tree and visualized the recombination regions. As a result, the 233 KL47 and KL64 CRKP strains could be divided into a total of five clades, with all KL47 CRKP located in one large clade and the closest related KL64 CRKP showing more recombinant nucleotide sequence mutations (Figure 5).

### 3.5. The Potential Mechanisms for the Evolution of ST11 CRKP from KL47 to KL64

We first compared the nucleotide sequence of KL47 and KL64 in the VR2 variable region and discovered that the KL64 strains possess more nucleotide sequences in this region, indicating that mutations occurred by increasing the number of nucleotide sequences in the VR2 variable region of the *wzc* gene based on KL47, facilitating the serotype shift. The nucleotide sequence profile of the *wzc* gene in the CD1-VR2-CD2 region directly affects the serotype (Figure 6A). The variation in the serotype-determining *wzc* gene is essentially the same in strains ST11-KL47 and ST11-KL64 of the same serotypes. We selected two strains with complete genome sequences in ST11-KL47 and ST11-KL64, which are highly representative and can provide more reliable results. Representative strains 573.19986 and 573.25802 were chosen for KL47 and KL64, respectively.

The region of KL47 compared to the *wzc* gene produced two sequences of 1491 base pairs (bp) and 342 bp, totaling 1835 bp, with a GC content of 53.84%, whereas the KL64 CRKP compared to the *wzc* gene produced a sequence of 2138 bp, with a GC content of 58.14% (Figure 6B). The nucleotide sequence length in the CD1-VR2-CD2 region of KL64 CRKP was increased by around 303 bp compared to KL47, and it had a greater GC content, which encouraged the mutation of the serotype from KL47 to KL64. After Prokka annotation, we discovered that the *cps* area contained a particular tyrosine-protein kinase (Putative tyrosine-protein kinase in *cps* region), leading us to believe that the recombination events took place at the Capsular biosynthesis site.

## 4. Discussion

Compared with previous studies [24,25,26], our global investigation of ST11 CRKP over 10 years demonstrated a gradual unification of the most prevalent clonal strains of carbapenem-resistant *K. pneumoniae* bearing *bla*_KPC-2_ in a dominant position. Blood was the predominant sample type in our collection of ST11 CRKP, while China was one of the main country sources of the strains, suggesting that the transmission of CRKP in the blood is common worldwide, especially in China, and requires more attention. In addition, *bla*_NDM-1_ and *bla*_OXA-48_ are significant carbapenemase genes in ST11 CRKP, while the clonal spread of *bla*_OXA-48_, *bla*_KPC-2,_ and *bla*_CTX-M-14_ was responsible for the outbreak of ST11 CRKP over a long period [27]. Most ST11 CRKP strains were found in China between 2011 and 2020 [28], demonstrating that serotypes of ST11-KL47 and ST11-KL64 are evolving and spreading more quickly there. Comparison to previous studies found that ST11-KL47 and ST11-KL64 CRKP have become dominant prevalent variants that cannot be disregarded and require attention. In our study, ST11-KL64 CRKP has been spreading worldwide and has begun to displace KL47 as the predominant serotype of ST11 CRKP, which was similar to a previous study [10]. KL47 could evolve into KL47 and KL64, a phenomenon known as transmission evolution [26]. It is undeniable that the evolution of KL47 to KL64 is gradually becoming the mainstream trend, and ST11-KL64 CRKP will account for an increasing proportion of all ST11 types. Additionally, KL64 is linked to high virulence [29], suggesting that ST11 carbapenem-resistant *K. pneumoniae* is evolving into a highly virulence strain.

Previous research has shown that the *rmpA*/*rmpA2*, *iucABCD,* and *iutA* genes were present in all ST11-KL64 CRKP. By examining 122 strains of ST11-KL64 CRKP, we found that the percentage of these genes carried varied between 36.89 and 50.82 percent when the Kleborate software identified the strain as ST11-KL64 CRKP. It is worth noting that ST11-KL64 strains carrying *rmpA*/*rmpA2*, *iucABCD*, and *iutA* genes all harbored the *bla*_KPC-2_ gene, and were only isolated from Chinese samples.

It was found that ST11-KL64 CRKP carried more virulence genes [median 78 (*P*_25_~*P*_75_: 72~79.25)] than KL47 [median 63 (*P*_25_~*P*_75_: 63~69)]. A comparison of the two serotypes regarding the number of 134 virulence genes carried revealed 35 virulence genes with statistically significant differences, including *rmpA*/*rmpA2*, *iucABCD*, *iutA* genes, etc., confirming that these virulence genes were differential genes between the two serotypes ST11-KL64 and ST11-KL47, consistent with previous studies [10]. Furthermore, it has been demonstrated that high virulence genes carrying KL47 and KL64 CR-hvKP exhibit a high level of clonality [30], confirming the evolutionary advantage of propagation when high virulence genes and carbapenem resistance genes co-exist.

There are variable numbers of virulence genes and plasmids in the ST11-KL47 and ST11-KL64 CRKP. The CD1-VR2-CD2 region of the *wzc* gene exhibits a nucleotide sequence increase of about 303 bp in KL64, as well as a better GC content, which facilitates the transformation from KL47 to KL64. The evolution of the serotype to KL64 likely resulted from mutations in various regions of the nucleotide sequence, and KL64 CRKP itself underwent recombination in specific regions during evolution, leading to the production of certain virulence genes such as *rmpA2*, *iucABCD,* etc., rather than the acquisition of more virulence genes or plasmids leading to the evolution of the serotype to KL64 [24].

Our research has certain limitations. Firstly, only the PATRIC database was used for sample screening, and there was insufficient diversity among the CRKP strains included in the study. Secondly, there were no related experiments performed to demonstrate the effect of recombination of particular region fragments on the transformation of KL47 into KL64. However, in this study, we characterized ST11 CRKP strains from across the globe. At the same time, the evolutionary mechanisms of the serotypes of ST11-KL47 and ST11-KL46 were studied in depth, which can be used as a reference for evolutionary studies of KL47 to KL64.

In summary, we found that KL47 and KL64 were the main serotypes of ST11 *K. pneumoniae*. KL47 and KL64 belong to a large evolutionary branch, and KL47 is the evolutionary ancestor of KL64. Comparative analysis of differential genes revealed that ST11-KL64 carries multiple virulence genes that might arise from a recombination process. In addition, KL64 CRKP carried 303 bp more nucleotide sequences in the CD1-VR2-CD2 region with higher GC content (58.14%) than KL47 CRKP, which facilitated the evolution of the serotype toward KL64. The discovery that mobile genetic elements play a role in the evolution of recombination provides a new basis for exploring the evolutionary process of the ST11-KL64 CRKP [10].

## Figures and Tables

**Figure 1 genes-13-01624-f001:**
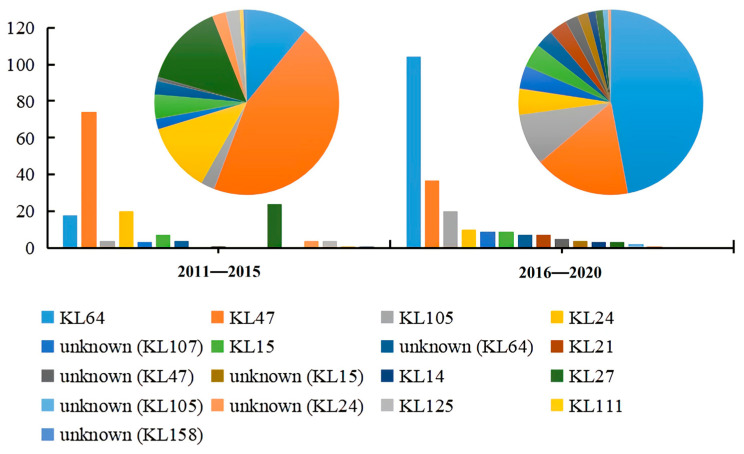
Serotype variation over 10 years. ST11 CRKP serotype distribution in 2011–2015; ST11 CRKP serotype distribution in 2016–2020. Unknown (KL64), an unknown serotype similar to KL64, and other unknown serotypes are indicated as well.

**Figure 2 genes-13-01624-f002:**
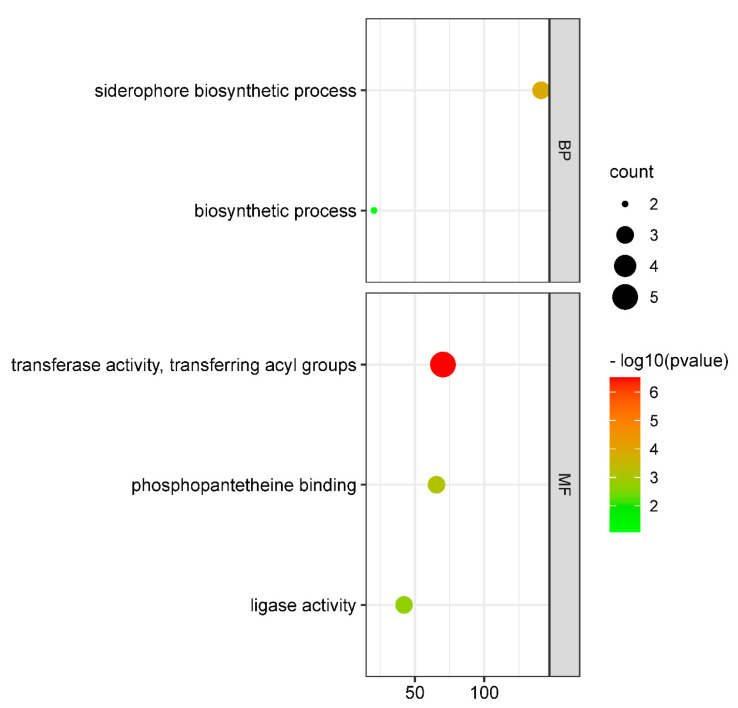
GO analysis of differential genes. Thirty-five differentially virulent genes were included in the GO enrichment analysis using DAVID software, and bubble plots were visualized using R v4.2.0.

**Figure 3 genes-13-01624-f003:**
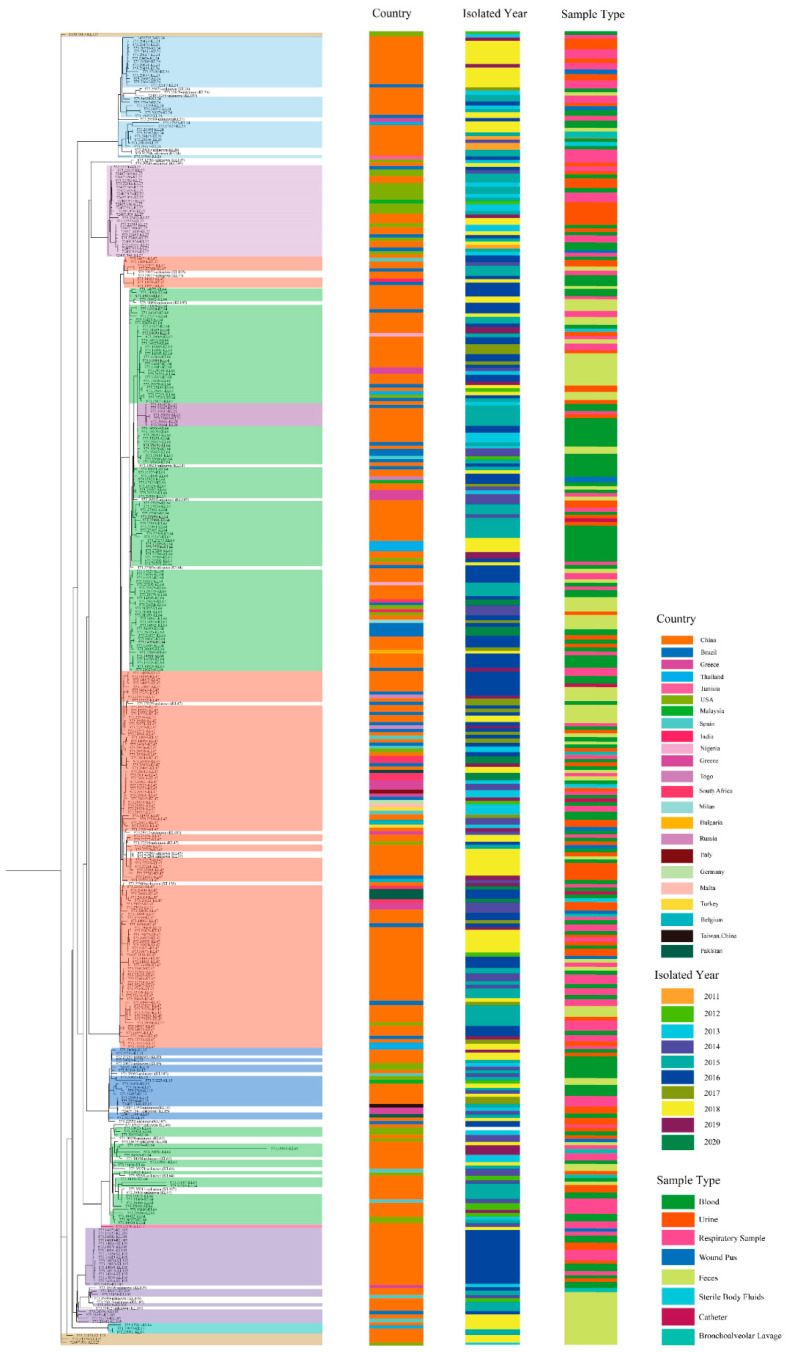
Phylogenetic tree of 386 ST11 CRKP strains. The phylogenetic tree is colored according to the different serotypes. The country of collection, year of collection, and sample type of each strain are annotated with color.

**Figure 4 genes-13-01624-f004:**
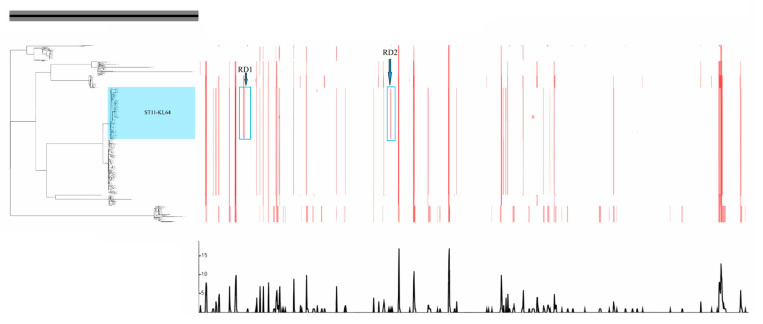
Recombination analysis of 386 ST11 CRKP isolates. Recombinant genomic regions (RD1 and RD2) were predicted by Gubbins and visualized by Phandango.

**Figure 5 genes-13-01624-f005:**
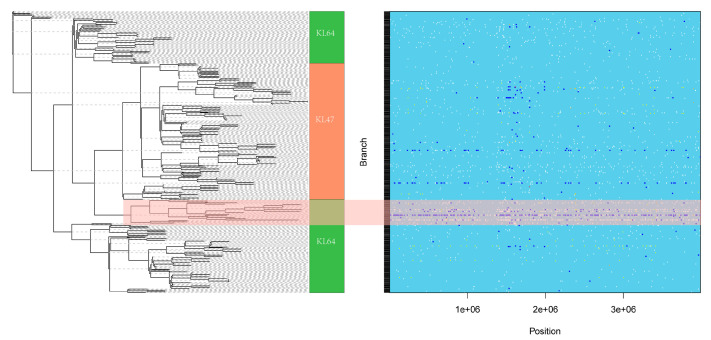
Phylogenetic and recombination analysis of KL47 and KL64. Images drawn with ClonalFrameML. The 233 strains of KL47 and KL64 CRKP can be divided into a total of 5 clades; KL47 CRKP all located in a large clade. The blue boxed area on the right side of the graph corresponds to the recombination of each strain in the region of positions 1 × 10^6^ to 3 × 10^6^.

**Figure 6 genes-13-01624-f006:**
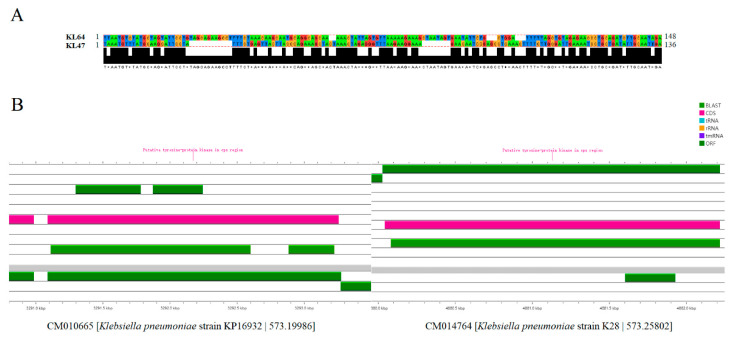
Serotype comparison of KL47 and KL64. (**A**) Baseline comparison of the CRKP VR2 variable region in KL47 and KL64; (**B**) Sequence characteristics of KL47 and KL64 in the wzc gene’s CD1-VR2-CD2 region. KL64 CRKP has more nucleotide sequences in the VR2 region. The region of KL47 alignment re-sults in two sequences of 1491 bp and 342 bp, respectively, with a total of 1835 bp and a GC content of 53.84%, while the KL64 CRKP obtained a sequence of 2138 bp after alignment with the wzc gene, with a GC content of 58.14%.

**Table 1 genes-13-01624-t001:** Comparison of differences in the number of resistance genes, virulence genes, and plasmids carried between KL47 and KL64 CRKP.

Variables	Group		*p*-Value
	KL47	KL64	
	*n* = 111	*n* = 122	
Number of Cabarpenemase genes	15 (14, 18)	15 (14, 18)	0.380
Number of plasmids	4 (3, 4)	3 (3, 4)	0.003
Number of virulence genes	63 (63, 69)	78 (72, 79.25)	<0.001

**Table 2 genes-13-01624-t002:** Differential of virulence genes between KL47 and KL64.

Virulence Genes	Group		χ^2^	*p*
	KL47	KL64		
	*n* = 111	*n* = 122		
*acrA*	105	122	4.786	0.029
*clbA*	1	19	15.946	<0.001
*clbB*/*clbC*/*clbD*/*clbE*/*clbF*/*clbG*/*clbH*	1	20	17.011	<0.001
*clbI*/*clbN*/*clbO*	1	19	15.946	<0.001
*clbL*/*clbM*/*clbP*/*clbQ*/*clbS*	1	20	17.011	<0.001
*entF*	111	113	6.647	0.010
*glf*	3	117	202.116	<0.001
*gnd*	3	122	221.262	<0.001
*iucA*	32	60	10.075	0.002
*iucB*	34	62	9.779	0.002
*iucC*/*iucD*/*iutA*	34	61	9.030	0.003
*mrkI*	101	121	8.665	0.003
*rmpA*	1	45	47.497	<0.001
*rmpA2*	30	59	11.205	0.001
*senB*	0	8	5.689	0.017
*vipA*/*tssB*	106	73	41.509	<0.001
*wbbM*	0	114	203.085	<0.001
*wbbN*	0	117	213.819	<0.001
*wbbO*	0	110	189.586	<0.001
*wzm*	0	118	217.522	<0.001
*wzt*	0	117	213.819	<0.001

## Data Availability

Genomic data for 386 ST11 CRKP downloaded from the Pathosystems Resource Integration Center (PATRIC) are shown in Appendix A.

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
