# Peer review of "Genomic Evolution of ST11 Carbapenem-Resistant Klebsiella pneumoniae from 2011 to 2020 Based on Data from the Pathosystems Resource Integration Center"

_genes, 2022, doi:10.3390/genes13091624_

Round 1

Reviewer 1 Report

Of what significance of the following virulence genes in the evolution wbbM  wbbN wbbO wzm wzt, knowing fully that these genes are completely absent in KL47 compared to their abundant in KL64?

Please comment on the uniqueness of these genes entF glf gnd 111 ,3 ,3 and 113 ,117, 122 as observed in KL47 and KL64 respectively please table 2 connection with hypervirulent nature of some strains of Kp.

It was said that more KL64 and KL47 belong to an evolutionary branch with higher homology (81.97 %, 100/122). which is the ancestor of three serotypes KL64, KL47 and KL21, as well as certain isolates of unknown serotype, was further related to KL27. This aspect needs further proof quantitatively or qualitatively from your result as I do not see where the justification is coming from here in the sentence except if trying to work to answers, based on your presented results.  Please clarify.

The authors chose one KL47 strain and one KL64 strain from the 386 ST11, CRKP strains for comparison of serotype differences, to examine information on the mutation of base sequences. Why choosing one and not more for precision? Considering the large number of KL47 and KL64

This is not a limitation as well.  Make your conclusion include take-home points and possible future research. Kindly separate your limitations from your conclusion which is expected to be derived from your findings. Find attached the track changes in pdf format.

Reviewer 2 Report

In this article, the authors did an excellent job of providing information on the genomic evolution of ST11 carbapenem-resistant klebsiella pneumoniae from 2011 to 2020 throughout the world using different statistical and bioinformatical analyses. Any bacterium showing resistance to the carbapenem group of antibiotics is a serious public health issue. I believe this study will assist the decision-makers in taking initiatives regarding antibiotic selection. However, I have a few major comments, which must be addressed before publication. Please check all the calculations carefully. I have found some discrepancies in the data. They must be addressed properly. Please check my comments below:

Major Comments

- Why did you show your results separately, e.g., 2011-2015 and 2016-2020? If there is any purpose, please mention it in the materials and methods.

-  “In total, blood samples accounted for 31.09 % (120/386) of all sample sources, while the proportions of feces, respiratory fluids, urine, wound pus, alveolar lavage fluid and catheter samples were 20.21 % (78/386), 23.06 % (89/386), 18.39 % (71/386), 2.85 % (11/386), 2.07 % (8/386) and 1.04 % (4/386), respectively”- Here, I think one sample type is missing, most probably sterile body fluid samples (maybe the number of this type sample will be five, because the overall number of samples is 381/386 as mentioned here). If you have any reason to omit this sample and the sample size, why? Also, please arrange the number of samples sequentially, e.g., 120>89>78>71>11>8>5>4. But you’ve used as 120>78>89>71>11>8>4.

- Please check the calculation of the data thoroughly. I have found some errors, e.g., 74/165 should be 44.85%, but you have written 45.1%. I couldn’t check all the data, please check all of them properly.

- “There were 111 KL47 CRKP, which were isolated from…..” and “Five kinds of carbapenemase genes were identified in the KL47 CRKP strains, including blaKPC-2 (95.54%, 107/112)……”? 111 or 112? Which one is correct? Please clarify it.

-  “There were 122 KL64 CRKP, which were isolated from blood (36.89%, 45/122), feces (27.87%, 34/122), respiratory tract (17.21%, 21/122), urine (14.75%, 18/122), catheter (1.64%, 2/122) and sterile bodily fluids (0.82%, 1/122)”- Please check this properly. The sum of the KL64 CRKP is 121, but you’re stating that the number is 122. Please clarify it.

- “The majority of the strains collected from 2012 to 2020 were from China……”- why 2012-2020? Why did you omit 2011? Please clarify it.

- “There were 122 KL64 CRKP, which were isolated from blood…”” and “Four carbapenemase genes were present in the KL64 CRKP strains, including blaKPC-2 (96.75%, 119/123)” 122 or 123? Which one is correct? Please clarify it.

- For Table 1, would you please check the outcomes of the analysis again? Here, you’re stating that the P25~P75 of the plasmid number are similar (3,4), but the p-value is showing that the variation is highly significant (here, p< 0.01). Please check the outcomes of the analysis again for all the data.

Minor Comments

- “As an important gram-negative bacteria…….”. It should be “…….Gram-negative bacterium……”

- Please write “K. pneumoniae” throughout the whole manuscript after its first use, except if the name is placed at the start of a sentence.

- “bubble plots were generated using R v4.0.2.”. As you didn’t use any references, please mention the developer, or company name of R software, as well as the city and country name.

- “The Chi-square test was used to compare the differences…..”. What kind of Chi-square test did you use here? Chi-square goodness of fit test or Chi-square test of independence? Please mention it here. Also, I think you had to do Fisher’s exact test for showing variations where the number of counts was less than 5. When one or more cell counts in a 2×2 table are less than 5, Fisher’s exact test is typically used instead of the Chi-square test of independence. If you did it, please mention it here. Also, please mention the level of significance for your significant value. I believe the level of significance of your analysis was 0.05 (p≤ 0.05).

- Please make unfirm the percentage data. Sometimes you have used one decimal character, sometimes two characters. Also, sometimes you have omitted the decimals. Please check it thoroughly.

Reviewer 3 Report

The work included retrieval of data from PATRIC database and its extensive analysis to understand the mechanism of evolution of particular drug resistance in K. pneumoniae. Overall, the study design is good, and the work has been planned well. With few changes the article can be accepted for publication.

The title should be modified to include the name of the database used.

The reason for selecting KL47 and KL64 should be given in the abstract, as well.

Minor language errors need to be corrected. Check the manuscript thoroughly.

Do not begin a sentence with a numeral.

Many abbreviations have been used. Although most of those are well-known to the researchers working in the field, yet, to make the article interesting for a broad audience, kindly provide full form where possible.

In introduction, write about origin of sequence types, particularly ST11.

Section 2.2: Replace ‘carriage of’ with ‘carrying’.

Section 3.2: Replace ‘carriage’ with ‘presence’

Instead of ‘conclusion’ write discussion. Conclusion should be brief and a standalone section.

A significant portion of the results describe about the origin of the strains with respect to the clinical sample etc, however, authors have not ‘discussed’ this aspect in the paper.

Comparing results with others’ findings and common accepted facts will make this article more interesting.

Round 2

Reviewer 2 Report

The authors have addressed all the comments. I recommend publishing this manuscript in its present form.

However, I have a minor comment.

Line 169: It still remains 112. Please change it to 111.